# Low-Carbohydrate (Ketogenic) Diet in Children with Obesity: Part 1—Diet Impact on Anthropometric Indicators and Indicators of Metabolic Syndrome and Insulin Resistance

**DOI:** 10.3390/diseases13040094

**Published:** 2025-03-25

**Authors:** Ivanka N. Paskaleva, Nartsis N. Kaleva, Teodora D. Dimcheva, Petya P. Markova, Ivan S. Ivanov

**Affiliations:** 1Department of Pediatrics “Prof. Dr. Ivan Andreev”, Faculty of Medicine, Medical University of Plovdiv, 4002 Plovdiv, Bulgaria; nartsis.kaleva@mu-plovdiv.bg (N.N.K.); petya.markova@mu-plovdiv.bg (P.P.M.); ivan.ivanov@mu-plovdiv.bg (I.S.I.); 2Department of Medical Informatics, Biostatistics and e-Learning, Faculty of Public Health, Medical University of Plovdiv, 4002 Plovdiv, Bulgaria; teodora.dimcheva@mu-plovdiv.bg

**Keywords:** ketogenic diet, obesity, children, insulin resistance, metabolic syndrome

## Abstract

Background: The ketogenic diet has been successfully used in the last 100 years in the treatment of epilepsy and other neurological disorders. In recent decades, it gained wider application in the treatment of obesity, metabolic syndrome, and type 2 diabetes. However, there have been only a few studies on its use in children with obesity and associated metabolic disorders. Objectives: To determine the clinical and metabolic effects of a well-formulated low-carbohydrate (ketogenic) diet in children with obesity. Methods: One hundred children with obesity and metabolic disorders underwent initial anthropometric, laboratory, and ultrasound examinations. They were placed on a well-formulated ketogenic diet and monitored for 4 months. The 58 patients who completed the study underwent follow-up examinations to assess the effects of the diet on anthropometric, clinical, and laboratory markers of metabolic syndrome and insulin resistance, cardiovascular risk factors, and certain hormone levels. Compliance with the diet, common difficulties in adhering to it, side effects, and positive changes in the patients’ health were analyzed. Results: At the end of the study, the average weight loss for the entire group was 6.45 kg, with a reduction in BMI of 3.12 kg/m^2^. Significant improvements were also observed in insulin resistance indicators, including fasting insulin levels, HOMA-IR index, QUICKI (*p* < 0.0001), and adiponectin (*p* = 0.04). The cases of hepatosteatosis decreased twofold, the number of patients with arterial hypertension was significantly reduced, as well as the number of children receiving antihypertensive therapy. Additionally, the number of patients meeting the criteria for metabolic syndrome decreased threefold. Conclusions: A well-formulated short-term ketogenic diet is effective in treating obesity, metabolic syndrome, and related comorbidities, and can be part of a comprehensive approach for these patients.

## 1. Introduction

In recent decades, we have witnessed an unprecedented global epidemic of obesity, metabolic syndrome, and type 2 diabetes—chronic diseases that increasingly affect even children from a very young age. These metabolic disorders underline a range of complications and diseases that worsen quality of life and shorten its duration.

According to the WHO, the prevalence of obesity (BMI ≥ 30 kg/m^2^) in the global population is expected to increase from 14% to 24% in the years 2020 to 2035 [1]. Data from 2022 for Europe indicate that obesity among the adult population is 59%, and one out of three children suffers from this problem [2]. The COVID-19 pandemic was another warning of the seriousness and social significance of obesity and its associated comorbidities [3]. Obese patients were prone to severe COVID-19 courses, frequent early and late complications, and a greater risk of fatal outcomes [4,5].

Scientific progress and the introduction of new research methods contribute to a more detailed understanding of the precise pathogenetic mechanisms of obesity and its numerous environmental and genetic predisposing factors. Dozens of medications are used to treat obesity, and new ones are introduced to regulate appetite, control weight, and improve metabolism. Surgical interventions, such as bariatric surgery, are increasingly applied even in childhood [6]. Despite this, we are far from controlling the epidemic rise of obesity, which is increasingly becoming the new “normality” in modern society.

In childhood, the treatment of obesity usually begins with a change in eating habits and physical activity, although there is still no consensus on the most appropriate diet for overweight and obese children [7,8,9].

The leading scientific hypothesis for the pathogenesis of obesity in recent decades is the energy balance theory [10,11]. According to this, the modern obesogenic environment, with its abundance of highly processed, high-energy foods, is the main reason many people systematically overeat and thus consume more energy than they expend. This imbalance is exacerbated by sedentary lifestyles. According to this model, the right approach to obesity is to reduce energy intake (“eat less”) and to increase energy expenditure (“move more”). While chronically elevated calorie intake undoubtedly plays a significant role in the pathogenesis of obesity, other major issues like the hormonal influence of different foods on the body, their biological effects, and the interactions with fat storage mechanisms [12,13].

Recently, the carbohydrate–insulin model of obesity has gained more and more supporters. According to it, obesity cannot be explained solely by an imbalance of calories. Attention is focused on food glycemic load and insulin response, with a subsequent cascade of metabolic and physiological changes [14,15].

The ketogenic diet (KD) has been used for over 100 years with success in the treatment of childhood epilepsy, and in the last 20 years, it has gained increasingly widespread popularity in the treatment of various neurological, metabolic, malignant, and other diseases [16]. There are numerous reports of its effectiveness in the treatment of adults with obesity, metabolic syndrome (MS), and type 2 diabetes [17,18,19,20,21]. The significant restriction of digestible carbohydrates in this diet predisposes to a reduced insulin response after meals and lowered basal and postprandial insulin levels. This favors weight loss and reduction in insulin resistance (IR), which is the basis of metabolic syndrome and many of its complications [22]. There are fewer similar studies in children, which is probably one of the reasons this diet is not offered to children with obesity, metabolic syndrome, and impaired metabolic health.

Therefore, we conducted a clinical study to estimate the clinical and metabolic effects of a “well-formulated KD” in children with obesity, metabolic syndrome, and impaired metabolic health. We tracked the impact of a 4-month KD on multiple anthropometric, clinical, biochemical, hormonal, and ultrasound indicators of patients, and determined compliance, side effects, and common difficulties when adhering to the KD. In this article, we present the results of the KD’s effect on anthropometric measures and indicators of insulin resistance, metabolic syndrome, and impaired metabolic health.

## 2. Materials and Methods

### 2.1. Participants Selection and Study Design

One hundred children aged 7 to 18 years were selected to participate in the study. They were admitted for investigations at the Department of Pediatrics of University Hospital “St. George”, Plovdiv between 2021 and 2023. Inclusion criteria were the presence of obesity (according to WHO criteria—BMI ≥ 2SD from the mean for gender and age) and at least one criterion for impaired metabolic health: abdominal obesity, impaired fasting blood glucose, primary arterial hypertension, dyslipidemia, hyperinsulinemia, hyperuricemia, hepatic steatosis, and polycystic ovarian syndrome. Exclusion criteria were proven adrenal gland dysfunction, pituitary pathology, congenital metabolic disease, treatment with medication causing IR, or contraindication for KD: familial (genetic) hypercholesterolemia, nephro- or cholelithiasis, and a history of pancreatitis.

After signing informed consent, patients underwent initial anthropometric, clinical, laboratory, and ultrasound examinations. Each patient received oral and written instructions for the diet, as well as an individual menu based on age and food preferences. The first 50 patients received also a “CareSens Dual” biosensor system to measure serum levels of blood sugar and beta-hydroxybutyrate and were instructed to measure these values once a week (in the morning before breakfast) for additional control of compliance with the diet. Patients were instructed to keep a food diary with daily recordings of the number of meals, the type of consumed food, and the approximate amount of protein-containing foods. The diary also included weekly weight measurements in the morning before meals. For patients with arterial hypertension (AH), daily measurements of arterial pressure after 15 min of rest were also required. Patients’ parents sent the aforementioned reports on diet, weight, arterial pressure, blood glucose, and beta-hydroxybutyrate on a weekly basis. After 4 months, patients underwent second clinical, laboratory, and ultrasound examinations to assess the effect of dietary intervention (Figure 1).

The only recommended medications during the diet treatment were antihypertensive drugs for children with severe AH at the pediatric cardiologist’s discretion and L-thyroxine for patients with hypothyroidism. None of the patients took insulin sensitizers during the intervention. Girls diagnosed with polycystic ovarian syndrome did not receive hormonal therapy. No requirements for a specific level of physical activity or active sports were set for families before starting the diet. It should also be noted that for a large portion of the patients, the 4-month follow-up period coincided with a period of strict isolation and remote education during the COVID-19 pandemic, which further limited their daily physical activity.

During the 4-month follow-up period, 41 patients dropped out of the study for various reasons (Figure 1), but as far as the research team is aware, not a single patient withdrew from participation due to side effects of the diet.

### 2.2. Characteristics of the Group of 58 Patients Who Completed the Study

The 58 patients who completed the study with follow-up assessments ranged in age from 8 to 18 years (mean 13.79 ± 0.34) at the start of the diet, including 35 (60.34%) boys and 23 (39.66%) girls. They were distributed into three age groups: 8 (13.79%) children in the age range 8–10 years, 25 (43.10%) children 11–15 years old, and 25 (43.10%) 16–18 years old (Table 1).

In addition to obesity, all children who completed the study had an abnormally elevated waist circumference (above the 90th percentile for their respective age and gender)—a marker of visceral obesity, and basal or postprandial hyperinsulinemia, defined as having more than a 5-fold increase in basal insulin, or insulin > 100 mUI/mL, recorded during the oral glucose tolerance test (OGTT) with measurements at 0, 30, 60, and 90 min.

At the start of the diet, 33 (56.89%) patients met the criteria for metabolic syndrome, while the remaining 25 (43.10%) children had one or more criteria for impaired metabolic health. Primary arterial hypertension was diagnosed in 27 (46.55%) children. Hepatic steatosis was found in 39 (67.24%) of the children. Eight (34.78%) female patients were diagnosed with polycystic ovary syndrome based on the Rotterdam consensus criteria [23]. Seven patients (12.07%) had Hashimoto’s autoimmune thyroiditis—two (3.44%) were euthyroid under treatment at the start of the diet, and five (8.62%) were newly diagnosed and euthyroid without treatment.

According to their compliance with the diet, the patients were divided into three groups—patients with good, moderate, and poor compliance. Our assessment was based on information provided by parents, reported menus in food diaries, and measured values of beta-hydroxybutyrate. Patients with good compliance (strictly adhering to the diet) numbered 26 (44.82%), those with moderate compliance (following the diet with some deviations and exceeding the recommended carbohydrate intake) numbered 15 (25.86%), and those with poor compliance (frequent violations of the diet and multiple consumptions of inappropriate foods) numbered 17 (29.31%).

### 2.3. Ketogenic Diet

The proposed diet is a “well-formulated ketogenic diet (WFKD),” in accordance with the guidelines of S. Phinney and J. Volek [24], with the following composition, structure, and recommendations:Carbohydrate intake: up to 40 g per day, divided into 10–13 g per meal.Protein intake: 1–1.5 g per kilogram of ideal body weight per day.Fat intake: enough to induce satiety without excessive consumption.Three to four meals per day: breakfast, lunch, dinner, and a small afternoon snack if hungry.Calorie counting is not necessary. Patients should eat until satisfied while adhering to the guidelines given in the individual menu. Skipping a meal is permissible if not hungry, but prolonged and intentional starvation is not recommended.Preference for natural foods: meat, fish, full-fat dairy products, eggs, low-carbohydrate vegetables, and a small number of low-carb fruits.Avoidance of processed, packaged foods, soft drinks, sweetened juices, and liquids.Fluid intake without sugar: 30–40 mL per kilogram per day.

Patients were advised to consume an adequate amount of protein (1–1.5 g/kg ideal body weight), distributed evenly among meals. Carbohydrates were restricted significantly, with recommendations to consume primarily low-carb vegetables, a small number of fruits with lower sugar content, and, if desired, to prepare low-carb substitutes for bread and desserts. Children were instructed to eat until satisfied with appropriate, preferably natural, unprocessed foods, without monitoring the caloric content of the food, but also without consuming excessively large amounts of fats (such as cream, tahini, mayonnaise, and butter).

### 2.4. Clinical Investigations

The focus of the clinical examination was on the following:Skin and its appendages (acanthosis nigricans, striae, acne, and the presence of increased hair in androgen-dependent areas in pubertal girls).Distribution of increased adipose tissue in different parts of the body.Cardiovascular system: measurement of heart rate and rhythm by auscultation; measurement of blood pressure (BP) with an age-appropriate sphygmomanometer under standard conditions. AH was diagnosed and classified according to the criteria of the European Society of Hypertension [25] based on historical data or BP measurements by parents and/or the research team.

### 2.5. Anthropometric Measurements Included

Height, weight, and waist circumference.Calculation of BMI and waist-to-height ratio.

### 2.6. Laboratory Investigations

Oral glucose tolerance test (OGTT) with measurement of blood glucose and insulin at 0, 30, 60, and 120 min.Complete blood count, glycated hemoglobin, analyzed on an automated hematological analyzer Advita 2120, Siemens Healthcare Diagnostics INC., Erlangen, Germany.Biochemical parameters: Lipid profile (total cholesterol, LDL cholesterol, HDL cholesterol, triglycerides), transaminases (ALT, AST), gamma-glutamyl transferase (GGT), urea, creatinine, uric acid, analyzed using original turbidimetric and immunoturbidimetric programs on a biochemical analyzer AU 480, Olympus; Beckman Coulter, Inc., Co Clare, Ireland.Hormones: Insulin, thyroid hormones (TSH, T3, T4), cortisol, testosterone, LH, FSH, analyzed using chemiluminescent immunoassay (CLIA) on an automated immunochemical analyzer Access 2, Beckman Coulter, Inc., Ireland.HOMA-IR and QUICKI indices were calculated (using online calculators) [26].Fatty liver index (FLI) was calculated [27].Urine analysis for semi-quantitative assessment of sugar and acetone; urine calcium/creatinine ratio (calculated UCa/UCr in mmol/L).Adiponectin level was measured using the BioVendor Human Adiponectin ELISA test.

### 2.7. Ultrasonographic Investigations

Echocardiogram.Ultrasound examination of the urinary tract system and liver.

### 2.8. Statistical Analysis

Continuous variables are expressed as mean and standard deviation (SD) in accordance with the assumption of normality, categorical variables are expressed as counts and percentages. To compare the differences in repeated measurements after the KD was applied, we used the Wilcoxon signed-rank test for non-normally distributed data and the paired *t*-test for normally distributed data. To assess the normality of the data distribution, we used the Shapiro–Wilk test. For categorical variables, we performed Mann–Whitney U tests to examine differences in diet compliance.

In the context of statistical analysis, values assigned to *p*-values < 0.05 were accepted as statistically significant. The effect sizes (Cohen’s d) were calculated for significant findings to evaluate the magnitude of the observed differences.

The robustness of the findings was subjected to further assessment through the execution of sensitivity analyses, which involved the exclusion of outliers. All statistical analyses were performed using IBM SPSS Statistics v.23 (Armonk, NY, USA).

## 3. Results

### 3.1. Anthropometric Indicators

After the 4-month dietary intervention, we observed a statistically significant decrease in all monitored anthropometric indicators related to obesity. The patients’ body weight decreased on average by 6.45 kg, with the reduction being significant in all three age groups (Appendix A).

There was also a statistically significant decrease in the mean BMI by 3.12 kg/m^2^ (*p* < 0.0001), waist circumference (*p* = 0.003), and waist-to-height ratio (*p* < 0.0001) (Table 2).

### 3.2. Indicators of Glucose Metabolism, Insulin Resistance, Impaired Metabolic Health, and Metabolic Syndrome

After completing the diet, there were no statistically significant changes in the mean levels of fasting glucose and glycated hemoglobin for the whole group. A significant decrease in fasting glycemia was reported only in the group of female patients when analyzing this indicator by gender (Appendix A). Significant changes were observed in parameters related to IR—a decrease in mean values of basal insulinemia (*p* < 0.0001) and HOMA-IR index (*p* < 0.0001), and an increase in QUICKI (*p* < 0.0001) and mean adiponectin levels (*p* = 0.04), (Table 2).

A statistically significant decrease was observed in mean triglyceride levels (*p* = 0.001), while HDL-C showed no statistically significant change at the end of the diet (*p* = 0.25).

During the 4-month follow-up period, we noted a significant decrease in the number of patients with elevated blood pressure. Prior to the diet, 27 (46.55%) children were diagnosed with AH, and 12 of them required antihypertensive therapy. At the end of the diet, only 11 (18.96%) children continued to have elevated blood pressure, and only 6 of them needed antihypertensive medication. Good compliance with the diet was associated with better final outcomes—in the group with good compliance, only one patient remained with high blood pressure, (Figure 2). At the end of the study, there was no significant change in mean uric acid levels (*p* = 0.37).

Before starting the diet, 39 (67.24%) of the children were diagnosed with non-alcoholic fatty liver disease (NAFLD) (based on ultrasound criteria), and the mean fatty liver index (FLI) for the entire group was 60.60%, confirming the high frequency of this disorder among our patients. Abnormally elevated liver enzymes (ALT, AST, or GGT) were found in ten patients (5.80%), and one patient had steatohepatitis.

After completing the diet, a significant decrease in the number of patients with NAFLD was observed (*p* < 0.0001). After the KD, only 20 (34.48%) children remained with ultrasound criteria for hepatic steatosis. The mean FLI decreased significantly to 40.67. A statistically significant decrease was observed in the mean levels of all investigated liver enzymes, although their mean levels remained within reference ranges. In nine out of ten patients with elevated liver enzymes before the KD, a decrease or normalization of their values was observed at the end of the follow-up period. A significant decrease in liver enzymes was also noted in the patient with steatohepatitis, despite his poor compliance with the diet in the last month.

At the end of the dietary intervention, the number of patients meeting the criteria for metabolic syndrome decreased more than threefold (*p* < 0.05).

### 3.3. Side Effects and Complaints

Side effects and complaints during the dietary intervention were observed in 27 (46.5%) of the patients, (Table 3). They were mostly mild, transient, more common in the first 2–3 weeks, and did not necessitate discontinuation of the diet in any of the children.

The most common complaint was headache—in 24% of the children (half of them had accompanying hypertension and episodic headaches before starting the KD). None of the patients reported hypoglycemia based on measurements with the provided glucometer. Fatigue was reported in 17% of the patients, mainly in the first month of starting the diet, which is not unusual for the adaptation period to the KD. Increased fluid intake and increasing the amount of food consumed had a good effect on most of them. Gastrointestinal complaints were reported in 15 (25.8%) of the patients. Seven (12%) had constipation, most commonly in the first 2–3 weeks after starting the diet. It was noticeable that these were mainly children with improper dietary habits before the diet, disliking almost all vegetables. These patients were advised to increase the amount of fiber consumed in the form of low-carbohydrate vegetables, as well as the amount of fluid intake. Four (6.88%) reported short episodes of diarrhea, and four (6.88%) reported abdominal pain—one of whom was a girl with dysmenorrhea. Only five (8.6%) of the children reported feeling hungry, more commonly in the first month of starting the diet. One of the girls (1.72%) reported a lengthening of the intervals between menstrual cycles.

Since classic KD is associated with an increased risk of nephrolithiasis, we also monitored calcium excretion and ultrasonographic characteristics of the excretory system in our patients. There was no statistically significant difference in mean serum creatinine levels (*p >* 0.05) and the calcium/creatinine ratio in urine (*p >* 0.05) before starting and after completing the diet. The ultrasound findings in the examination of the urinary system also remained unchanged in all observed patients.

### 3.4. Sensitivity Analysis

The exclusion of outliers (values > 2.5 SD from the mean) did not materially alter the findings of our analysis. This supports the robustness of the results, indicating that the conclusions drawn are reliable and not dependent on these extreme data points. By ensuring that the results are consistent with and without the outliers, we have enhanced the credibility of our analysis.

### 3.5. Effect Sizes

The effect sizes (Cohen’s d) for statistically significant differences between groups are moderate to large, ranging from 0.5 to 0.92. This provides evidence that the observed differences are meaningful and impactful.

## 4. Discussion

In our study, we investigated the clinical and metabolic effects of a “well-formulated KD” in children with obesity, MS, and impaired metabolic health. At the end of the 4-month dietary intervention, our patients managed to significantly reduce their weight. However, childhood is characterized by uneven growth and development and weight loss is not the most accurate indicator of fat loss. Due to significant variations among our patients in terms of age (ranging from 8 to 18 years) and initial body weight (ranging from 43 kg to 146 kg), we also tracked other anthropometric indicators that more precisely reflect weight loss (BMI and percentage of weight loss) and analyzed the results separately in different age groups.

Children from all age groups reduced significantly their BMI and had a statistically significant percentage of weight loss, which was highest (−8.2%) in patients aged 8 to 10 years (Appendix A). There was also a significant reduction in the indicators reflecting visceral obesity—waist circumference (*p* < 0.003) and the waist-to-height ratio (*p* < 0.0001). It is known that visceral adipose tissue has different metabolic characteristics compared to subcutaneous adipose tissue. Visceral obesity is clearly associated with an increased risk of cardiometabolic diseases, disturbances in glucose and lipid metabolism, predisposition to some malignant diseases, increased susceptibility to infections, and infectious complications, among others [28,29,30,31,32,33].

Large studies have shown that the distribution of body fat has a genetic basis [34,35] and targeted genetic studies among obese patients in the future would likely contribute to more precise identification of individuals predisposed to visceral fat accumulation and, accordingly, most at risk of developing cardiometabolic complications.

Genetic testing of some genetic variants (like rs1421085 T>C in the FTO gene) [36] can help in crafting a personalized dietary approach, providing targeted advice for the individual depending on their ability to better metabolize fats or carbohydrates.

The effectiveness of the KD in reducing body weight has been demonstrated in numerous published studies, systematic reviews, and meta-analyses, the majority of which were conducted in adult patients [17,18,19,20,21]. There are few similar studies on children. In a 6-month study comparing the effects of KD to a low-fat, hypocaloric diet in two groups of obese children, Partsalaki et al. reported weight loss averaging 8 kg and a decrease in BMI of—3.7 kg/m^2^ in KD patients [37]. In another study involving children aged 5–18 years, Pauley et al. monitored changes in weight and BMI in 130 children over 3–4 months while adhering to a KD with carbohydrate restriction < 30 g/day and recommending children to eat until satiety without calorie counting. During the specified period, the patients lost an average of 5.1 kg, with a mean reduction in BMI of 2.5 kg/m^2^ [38]. The results of our patients are like those reported in other studies of the KD in children with obesity.

It is known that in obesity, signs of IR and metabolic disorders are often found, even if the patients do not meet all the criteria for MS. This has led to the introduction of the concepts of “metabolically healthy/unhealthy obese” [39,40]. It is considered that this division is particularly useful in childhood, where full-blown MS is diagnosed less frequently, but the symptoms of impaired metabolic health in children with obesity predict an increased risk of future chronic diseases and complications. These data explain why we tracked the effect of KD on the key indicators of MS (fasting glycemia, triglycerides, HDL-C, AH), as well as other markers of impaired metabolic health—fasting insulin levels, IR indices (HOMA-IR, QUICKI), adiponectin, uric acid, NAFLD, etc.

We did not observe significant changes in fasting glycemia and mean levels of glycated hemoglobin in the entire group of patients after completing the diet, which was expected since in children with obesity these indicators are most commonly within reference values. We associated the statistically significant decrease in fasting glycemia in female patients with better compliance with the diet in this group (Appendix A).

However, the indicators reflecting insulin sensitivity—HOMA-IR and QUICKI—were significantly improved in our study. So far, there are no universally accepted reference values for these indices, as well as for basal and postprandial insulin levels in childhood. The main reason for this is likely the fact that puberty is a period of mild physiological IR, which is associated with characteristic hormonal changes along the growth hormone/IGF-1 axis. In a study of children using a hyperinsulinemic-euglycemic clamp, Moran et al. found that some IR occurs shortly after the onset of puberty (Tanner stage II), peaks in Tanner stage III, and by the end of puberty (Tanner stage V), insulin sensitivity returns to pre-pubertal levels [41].

The initial data of our patients—average basal insulin levels of 20.12 ± 8.88 mIU/L, HOMA-IR of 4.51 ± 2.20, QUICKI of 0.31 ± 0.019, and Matsuda of 2.32 ± 17.08—indicate undeniably the presence of IR exceeding physiological levels for children in puberty. All these indicators were significantly improved at the end of the study: mean fasting insulinemia—13.17 ± 5.81 mIU/L, HOMA-IR—2.85 ± 1.45, and QUICKI—0.33 ± 0.024. A significant increase was also observed in the average adiponectin level, confirming the improved insulin sensitivity of the studied children at the end of the diet.

Hyperinsulinemia—basal or postprandial—is considered by many authors to be a particularly important prerequisite for the development of chronic IR, MS, and associated disorders and diseases [42,43,44,45]. Several authors report in their studies on the effectiveness of the KD in lowering basal insulinemia [46,47,48]. Similar to our results, improved IR indices were also reported by the authors of all available studies that investigate the effects of a KD in children with obesity and MS [37,38,49,50].

In our study, after completing the KD, we performed OGTT in only one patient in whom basal insulin was within normal values in both tests. After the KD, a dramatic decrease in postprandial insulin levels was observed in OGTT. We suggest that this improvement might be true for the other patients with obesity and normal fasting insulinemia.

The average levels of total cholesterol, LDL-C, and HDL-C in our patients remained with no statistically significant change after KD. One of the main criteria for MS is elevated triglyceride levels, which are closely associated with atherogenic dyslipidemia and cardiometabolic risk. Elevated triglyceride levels are often found in patients with hepatic insulin resistance [51]. The significant reduction in this parameter in our patients, together with the decreased HOMA index, indicates improved hepatic insulin sensitivity, an important condition for normalizing overall metabolism.

Non-alcoholic fatty liver disease (NAFLD) is a disorder that occurs with high frequency in conditions where IR is present (obesity, type 2 diabetes), and hyperinsulinemia and increased de novo lipogenesis are almost universally found in patients with NAFLD [52].

The favorable effect of the KD on NAFLD is reported by many other authors. In a pilot study from 2007, Tendler et al., using liver biopsy, demonstrated the positive effect of the KD on hepatic steatosis in five obese patients [53]. In a study of the effect of the KD on NAFLD in obese patients, Luukkonen et al. found a pronounced reduction in hepatic insulin resistance, regardless of the increase in circulating non-esterified fatty acids [54]. D’Abbondanza et al. also reported a beneficial effect of the KD on hepatic steatosis in their obese patients [55]. In a randomized study in children with obesity and confirmed hepatic steatosis, Goss et al. found that for a short period of 8 weeks, the ketogenic diet led to a significant reduction in hepatic lipid content and emphasized the importance of restricting intake of sugar, glucose, and fructose to achieve such an effect [56]. In our study, we proved that a 4-month dietary intervention leads to almost a twofold decrease in the cases with hepatic steatosis. We noted a statistically significant decrease in ALT and GGT—enzymes whose elevation is usually observed in hepatic steatosis. The mean values of FLI were also reduced from 60.6 to 40.67.

To our knowledge, the effect of the KD on AH in obese patients is studied mostly in adults and the results are equivocal. In the meta-analysis by Castellana et al., improvement in systolic and diastolic blood pressure values was reported in patients after following a KD [57]. Another meta-analysis of randomized studies on the effect of the KD (Hession et al.) did not find a statistically significant change in blood pressure values in patients with obesity and accompanying comorbidities [58]. Cicero et al. reported a significant reduction in systolic blood pressure in their study of 377 patients with hypertension, type 2 diabetes, and dyslipidemia following a strict KD for only 3 months [59]. Barrea et al. reported a significant reduction in blood pressure, along with improvements in anthropometric parameters, body composition, and markers of systemic inflammation in women with obesity and AH who followed a low-calorie KD for 45 days. They suggest that the KD should be offered to properly selected and motivated patients suffering from obesity and hypertension since it may lead to discontinuation of antihypertensive drug therapy and remission of the disease [60].

The good results of our study regarding AH are likely influenced by the fact that our patients are children, assuming recently developed hypertension, and some of them had only mildly to moderately elevated blood pressure values. We admit that some of the elevated blood pressure measurements during the clinical examinations were “white coat hypertension” and that blood pressure might not be measured accurately at home. Nevertheless, we deem that the KD has led to normalization of blood pressure and discontinuation of antihypertensive therapy in a significant portion of our patients.

Although elevated levels of uric acid are not among the commonly accepted criteria for diagnosing MS, studies consistently find that hyperuricemia is often present in patients with obesity, MS, and type 2 diabetes, including in childhood [61]. It is reasonable to question the effect of the KD on uric acid levels, especially in patients with high initial values, as some elevation of serum uric acid is considered a typical marker of the several-month period of keto adaptation [24]. The lack of a statistically significant change in uricemia in our patients after the diet is likely indicative of achieved keto-adaptation.

Several authors have found the KD to be an effective approach for the management and reversal of metabolic syndrome [22,62,63,64].

Undoubtedly, among the most important effects of the KD that we observed in our patients was the reversal of MS. After completing the diet, the number of patients meeting the criteria for MS decreased by more than threefold, clearly demonstrating the effectiveness of the diet in overcoming IR and MS.

Compliance with a given diet is of paramount importance for its effectiveness. In our study, out of the initial 100 patients, 23 decided not to start the proposed diet at all. We assume that insufficient conviction within the family about the impact of diet on the child’s health problems (e.g., high blood pressure, liver steatosis, etc.) played an important role in this decision. It is also likely that some families had no genuine intention of making serious dietary changes, but joined the study primarily because of the opportunity for free, comprehensive initial examinations and consultations with various specialists (pediatric cardiologist, pediatric endocrinologist, ultrasound examination of abdominal organs), which are difficult to arrange within a short period in outpatient settings.

We also hypothesize that the dropout of some patients was due to the inability to attend follow-up visits at the clinic during the monitoring period, owing to the strict COVID-19 restrictions in Bulgaria.

A significant percentage of dropouts during follow-up has been reported not only in our study but also in numerous publications by other authors on the dietary interventions in children for periods longer than four weeks—ranging from 30% to 58% [37,38,49,50].

Of the 58 children who completed the study, 41 (70.69%) demonstrated good or moderate compliance with the diet. The group of children with good compliance showed the most significant weight reduction and the greatest improvement in terms of arterial hypertension, liver steatosis, and monitored laboratory indicators. Patients with moderate compliance also achieved good results, many of which were close to those of the patients with good compliance. This gives us reason to believe that even a moderate carbohydrate restriction (not in the ketogenic zone) in childhood would have a satisfactory effect on obesity, while at the same time improving compliance with the diet.

### Limitations and Issues of Concern

The four-month observation period defines our study as relatively short-term, so it cannot answer the question of whether the observed trend toward weight loss and other positive changes in the patients’ health will persist after longer periods of time. It is highly probable that if our patients return to their previous dietary habits (consumption of large amounts of sugary and starchy foods, carbonated and sweetened beverages, etc.), this will inevitably result in rapid weight gain and exacerbation or emergence of obesity-related disorders. However, one of our aims was also to educate patients about healthier eating habits, with a preference for natural, minimally processed, and micronutrient-rich foods, as well as significant restrictions on products from the confectionery industry and highly processed foods. These dietary habits will be beneficial regarding obesity and MS even without KD [65].

Another important, but unsolved, question is how long significant carbohydrate restriction can and should continue in children with obesity and MS. Patients with neurological disorders such as epilepsy usually remain on a KD for years, and in some congenital metabolic disorders (pyruvate dehydrogenase deficiency, Glut-1 deficiency), a lifelong KD is mandatory [66,67]. We believe that for most pediatric patients with obesity, such long periods of severe carbohydrate restriction are hardly necessary. After completing the four-month dietary intervention, our approach to patients was individualized and determined by the initial degree of obesity, the presence of accompanying health problems such as MS, AH, polycystic ovary syndrome, NAFLD, and our assessment of their improvement during the dietary intervention. For some children, a 4–5-month diet with significant carbohydrate restriction was sufficient to achieve normal or close to normal weight, as well as to normalize blood pressure, and some abnormal laboratory parameters. For these patients, we recommended a slow and gradual increase in carbohydrate intake by adding fruits, potatoes, or rice (depending on the child’s taste preferences) while simultaneously controlling weight. Some of these patients, with whom we still have contact, have maintained their achieved weight and feel well more than 1 year after completing the diet. Some children have further reduced their weight, while two others have started gaining weight again by returning to their usual unhealthy eating habits. For patients with severe obesity, serious metabolic disorders, and diseases such as severe hypertension, polycystic ovary syndrome, etc., we recommended prolonged feeding with significant carbohydrate restriction. Some of these children are still being periodically monitored in the clinic.

We did not include a control group of patients following another diet in our study to compare the results. There are publications in the literature about the successful implementation of other diets in children—for example, hypocaloric diet with fat restriction, etc. [37,49,58]. The aim of our study was to investigate the effects of a low-carbohydrate diet and to what extent it can be used in clinical practice in patients with the described health problems. Future long-term studies are undoubtedly needed to determine its efficacy and safety, especially in younger children.

## 5. Conclusions

A relatively short-term (4-month) “well-formulated KD” in children aged 8–18 years with obesity, IR, and MS leads to significant weight loss, improved insulin sensitivity, and regression of MS in a significant portion of the patients. We found that the KD had particularly good effects regarding primary AH and NAFLD. Most patients had good or moderate compliance with the diet, which was associated with a more significant improvement in many anthropometric and biochemical indicators. The reported complaints and side effects during the diet were mild and manageable and did not necessitate discontinuation of the diet. We believe that a low-carbohydrate diet can be used as one of the possible dietary options in a comprehensive approach in children with obesity and metabolic syndrome.

## Figures and Tables

**Figure 1 diseases-13-00094-f001:**
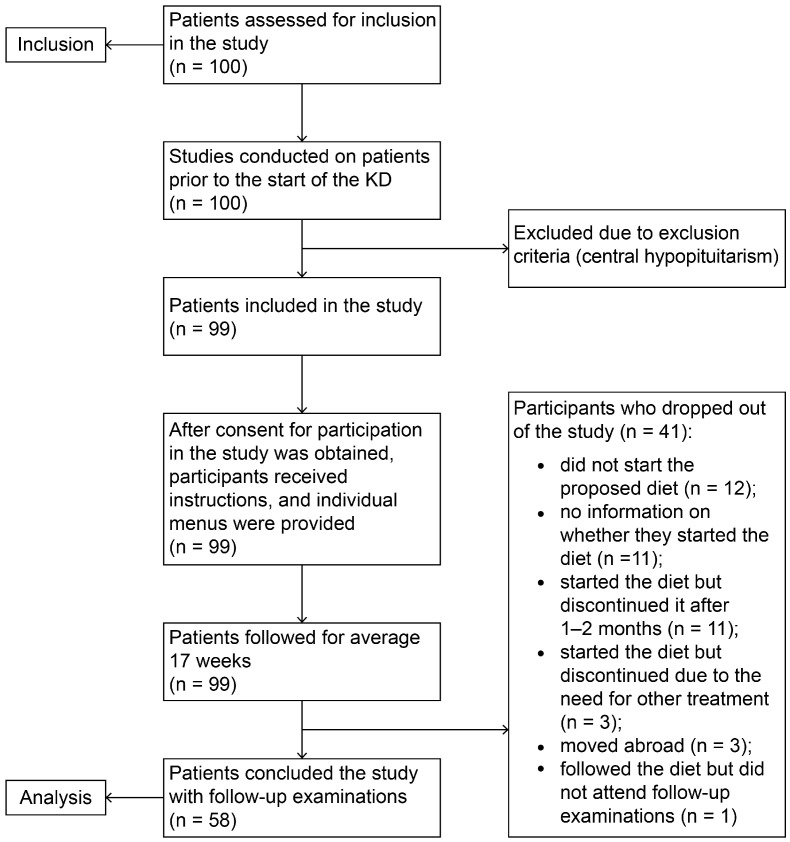
Flowchart of the study protocol.

**Figure 2 diseases-13-00094-f002:**
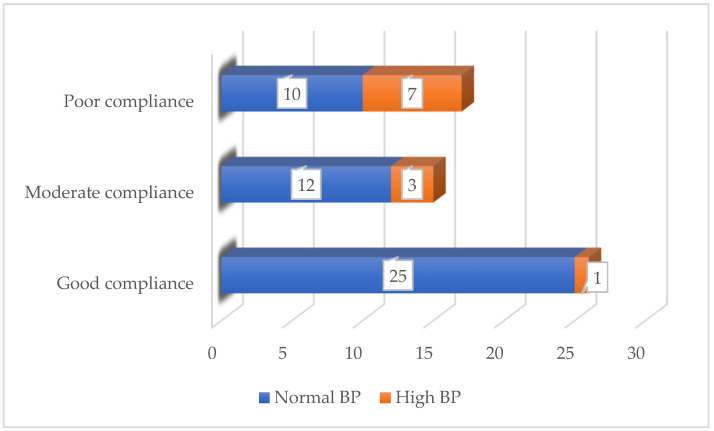
Patients with high BP after the KD according to compliance with the diet.

**Table 1 diseases-13-00094-t001:** Characteristics of the study group.

Variable	N = 58
Age, years (x ± SD)	(13.79 ± 2.63)
Age groups, n (%)	
8–10 years	8 (13.79)
11–15 years	25 (43.10)
16–18 years	25 (43.10)
Gender, n (%)	
Female	23 (39.66)
Male	35 (60.34)
Comorbidities, n (%)	
Metabolic syndrome	33 (56.89%)
One or more criteria for impaired metabolic health	25 (43.10%)
Polycystic ovary syndrome	8 (34.78%)
Primary arterial hypertension Hepatic steatosis	27 (46.55%) 39 (67.24%)
Hashimoto’s autoimmune thyroiditis	7 (12.07%)
Compliance with the diet, n (%)	
Good	26 (44,83)
Moderate	15 (25,86)
Poor	17 (29,31)

x—mean; SD—standard deviation.

**Table 2 diseases-13-00094-t002:** Anthropometric and laboratory parameters of patients who completed the study.

Variable		Before KD (n = 58)	After KD (n = 58)	*p*-Value
Anthropometric indicators	Weight, kg	89.43 ± 3.30	82.98 ± 3.25	<0.0001
	BMI, kg/m^2^	33.35 ± 7.29	30.23 ± 7.14	<0.0001
	Waist circumference, cm	103.97 ± 17.17	91.38 ± 15.03	0.003
	Waist-to-hip ratio	0.64 ± 0.09	0.55 ± 0.08	<0.0001
Laboratory indicators	Fasting glucose (mmol/L)	4.94 ± 0,45	4.80 ± 0,45	0.07
	Fasting insulin (mlU/L)	20.12 ± 8.88	13.17 ± 5.81	<0.0001
	HOMA-IR	4.51 ± 2.20	2.85 ± 1.45	<0.0001
	QUICKI	0.31 ± 0.019	0.33 ± 0.024	<0.0001
	Glycated hemoglobin (HbA1c) %	5.21 ± 0.37	5.30 ± 0.38	0.27
	Total cholesterol (mmol/L)	4.38 ± 0.76	4.26 ± 0.91	0.25
	HDL (mmol/L)	1.17 ± 0.21	1.15 ± 0.22	0.45
	LDL (mmol/L)	2.739 ± 0.68	2.742 ± 0.75	0.97
	Triglycerides (mmol/L)	1.07 ± 0.45	0.88 ± 0.36	0.001
	Triglycerides/HDL	2.22 ± 1.13	1.83 ± 0.87	<0.0001
	Creatinine (mmol/L)	63.48 ± 7.53	63.40 ± 10.19	0.924
	Uric acid (µmol/L)	380.59 ± 90.91	373.36 ± 91.65	0.37
	Urinary calcium/creatinine ratio (mmol/mmol)	0.195 ± 8.88	0.193 ± 5.81	0.71
	ALT (IU/L)	28.87 ± 35.49	22.62 ± 16.21	0.001
	AST (IU/L)	25.82 ± 12.86	23.30 ± 8.54	0.006
	GGT (IU/L)	24.17 ± 12.93	19.62 ± 8.72	<0.0001
	FLI	60.60 ± 29.45	40.67 ± 31.91	<0.0001
	Adiponectin (mcg/mL)	8.61 ± 3.61	9.13 ± 3.86	0.04

Data are presented as mean ± standard deviation (SD). Wilcoxon signed-rank test was used to assess change within groups. Mann–Whitney U test was used to compare change between groups when data were not normally distributed. *p* < 0.05 was considered significant.

**Table 3 diseases-13-00094-t003:** Side effects of the KD.

Reported Complaints and Symptoms During the Diet	Number of Patients	%
Constipation	7	12.04
Diarrhea	4	6.88
Abdominal pains	4	6.88
Fatigue	10	17.20
Hunger	5	8.60
Headache	14	24.08
Longer menstrual cycles (girls)	1	1.72
Change in breath	1	1.72
Nervousness	1	1.72

## Data Availability

Data will be made available upon reasonable request to the corresponding author due to privacy restrictions.

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
