# Peer review of "Low-Carbohydrate (Ketogenic) Diet in Children with Obesity: Part 1—Diet Impact on Anthropometric Indicators and Indicators of Metabolic Syndrome and Insulin Resistance"

_diseases, 2025, doi:10.3390/diseases13040094_

Round 1

Reviewer 1 Report

Comments and Suggestions for Authors

The aim of this study is to determine the clinical and metabolic effects of a well-formulated low-carbohydrate (ketogenic) diet in children with obesity.

Introduction provides sufficient background and includes relevant references. The study design is appropriate but given that the subject of the study is directly related to children and is highly controversial, It is difficult to accept a study whose “n” for children aged 8 to 10 years is 8.

As children, they also need to learn how to eat and maintain long-term dietary patterns that do not cause imbalances in their bodies.

In addition, the authors state that of the total number of patients, 58, there are 18 with low adherence to KD. This leaves a total of 40. Of these 40, it is not known how many are in the 8-10 age group. Therefore, the final number in this age group may be even lower.

For the other two age groups (11-15 years) and (16-18 years) there is no gender breakdown. In these age groups there may already be differences due to hormonal variations.

As a example Line 258-261…..Prior to the diet, 27 (46.55%) children were diagnosed with AH, and 12 of them required antihypertensive therapy. At the end of the diet, only 12 (20.68%) children continued to have elevated blood pressure, and only 6 of them needed antihypertensive medication. The percentages are for 58 people, including the 18 who were poor compliance to KD, i.e. who did not follow the diet at all. There is no indication of the age subgroup for which the results are valid.

This is even more important in the case of side effects and complaints,

Line 285-286 Side effects and complaints during the dietary intervention were observed in 27 (46,5%) of the patients, (table 3).

This is 27 out of 58 patients (46.5%). It is important to know whether these results are related to adherence to the diet. However, if we subtract the 18 patients with poor adherence, the effects would be greater. It would be expected that the greater the adherence to the diet, the greater the side effects.

There is clear evidence that the diet is leading to water and electrolyte imbalances and dehydration.

 In this sense, the data presented would be a good start as preliminary data, but I believe that a larger number of patients and their breakdown by age and gender is needed to increase the power of the results.

Reviewer 2 Report

Comments and Suggestions for Authors

Successful Ketogenic diets need to continue high fat intake without increasing saturated fat intake, This is usually done with monounsaturated fats such as olive oil etc.  Another important component is to maintain the high fibre intake . This will change the gut bacterial composition and maintain or increase the supply essential short chain fatty acids. It will also maintain the gut barrier (preventing leaky gut toxicity and activation of the innate immune system) to lipopolysaccharides and other inflammatory components from the gut to the circulation (Shown by the non elevated uric acid levels in the patients) In many studies there was a  reduction in the fibre intake and this changed those outcome of those studies considerably so this is an big factor in the success of the present study

               The absence of a rise in cholesterol also indicates compliance with the diet and consumption of the low cholesterol protein component of the diet.

The study has a compliant group , a partially compliant group and a non compliant group but no control group for comparison.  However the non compliant group could have served as a de facto control group had the analysis been done by splitting the results into the three groups of the study. This would also have changed the statistics as the non compliant group was included in table 2. The numbers of each group would have been smaller but still able to show significant changes.

Monitoring ketones at home was a good control for compliance as the dietary history and food diaries are notoriously inaccurate. The choice of oils for the study my have had some impact as a recent study in the Journal of Clinical Endocrinology and metabolism (2024 109: 2847-2856) has shown that fish oil had a worse cardiovascular outcome when compared with omega 3 fatty acids from vegetable source (ALA and DFA vs EPA)

The study is well done and provides significant information for the treatment of obese children.

However the impact may have been greater had the analysis been done by splitting the compliance of the participants. Significant changes were shown in all parameters analysed so the results will be even more encouraging.

Gastrointestinal effects are to be expected especially as some participants were reported to have vegetable poor diets prior to the study

Reviewer 3 Report

Comments and Suggestions for Authors

The study is ambitious and well executed, manuscript fine in all parts. I miss some points in the Discussion.

1. Genetics - see references in the enclosed wordfile - please add a paragraph in the discussion.

2. Brown fat - see reference in wordfile - this could have great effects - please discuss.

3. Surgery and drug therapy are best choices right now, diet in the long run even together with physical activity (there are scores - could have been included) are often not successful. See reference in enclosed wordfile.

4. Lenght of follow-up period. In limitation you mention that 4 months is a short period, what is the usual study period in metabolic/diet studies? Did you follow up your patients/voluntaries - did the weight reduction vanish after an extra 4 months period?

5. Puberty. 11-15 years, how many? In the younger group - had any started puberty?

6. Did you consider measuring subcutaneous and visceral fat with scanning or CT - these are more direct methods than BMI and waistline measurements. There are other methods to evaluate subcutaneous fat - please discuss and add a paragraph in Discussion.

7. Adiponectin. Why did you not measure leptin?

8. You screened for fatty liver, with ultrasound: you could have measured visceral and subcutaneous fat concomitantly.

Schleinitz, D., Böttcher, Y., Blüher, M. et al. The genetics of fat distribution. Diabetologia 57, 1276–1286 (2014).

Pulit SL, Stoneman C, Morris AP, Wood AR, Glastonbury CA, Tyrrell J, Yengo L, Ferreira T, Marouli E, Ji Y, Yang J, Jones S, Beaumont R, Croteau-Chonka DC, Winkler TW; GIANT Consortium; Hattersley AT, Loos RJF, Hirschhorn JN, Visscher PM, Frayling TM, Yaghootkar H, Lindgren CM. Meta-analysis of genome-wide association studies for body fat distribution in 694 649 individuals of European ancestry. Hum Mol Genet. 2019 Jan 1;28(1):166-174.

Zhang Z, Chen N, Yin N, Liu R, He Y, Li D, Tong M, Gao A, Lu P, Zhao Y, Li H, Zhang J, Zhang D, Gu W, Hong J, Wang W, Qi L, Ning G, Wang J. The rs1421085 variant within FTO promotes brown fat thermogenesis. Nat Metab. 2023 Aug;5(8):1337-1351. doi: 10.1038/s42255-023-00847-2. Epub 2023 Jul 17. PMID: 37460841.

Kelly AS, Armstrong SC, Michalsky MP, Fox CK. Obesity in Adolescents: A Review. JAMA. 2024 Sep 3;332(9):738-748. doi: 10.1001/jama.2024.11809. PMID: 39102244.

Reviewer 4 Report

Comments and Suggestions for Authors

The manuscript by Paskaleva et al. entitled “Low-Carbohydrate (Ketogenic) Diet in Children with Obesity: Part 1 – Diet Impact on Anthropometric Indicators and Indicators of Metabolic Syndrome and Insulin Resistance” was intended to investigate the clinical and metabolic effects of a well-formulated low-carbohydrate (ketogenic) diet in 16 children with obesity.

If we consider the social, health, economic and quality of life implications related to the presence of obesity in individuals, and even more so in children, we can consider the topic to be of great relevance and interest.

However, I believe that the study presents some important critical issues.

1)      I find that the age range chosen by the authors is too broad. Putting together the anthropometric, metabolic and hormonal characteristics (just to name a few) of pre- and post-puberty individuals is physiologically meaningless. In fact, the physical, psychological and especially hormonal transformations that characterize them are too different from each other. It would be like trying to put together the degree of acidity of an unripe fruit with a ripe one and evaluate how their mixed juice is perceived. It is clear that, if analyzed separately, the perception of the acidic fruit will be different from that the juice of the ripe fruit. Therefore, I expect that the responses to the DK of a child will be different from those of a teenager.

2)      Considering PCOS among the factors is wrong, if females are not separated from males in the analyses. In fact, it is a condition that affects only females. This further complicates the endocrine differences

3)      I assume that since the participants were children, the informed consent was signed by a parent or legal guardian.

4)      How were taste preferences determined?

5)      When keeping a food diary, reporting the number of meals and the type of food is not sufficient if the quantities are not indicated.

6)      L357-360: Are there any longitudinal studies that report data along these lines? If so, the authors should cite them. If not, they should remain more speculative.

7)      Conclusions: the results, and consequently the conclusions, are distorted by how the authors used the panel they had available. The analyses should all be redone, taking into account the age and sex of the participants more carefully.

Round 2

Reviewer 4 Report

Comments and Suggestions for Authors

1) I have checked the manuscript again and once again I do not find the figures and data separated by age and sex. There are two explanations: I have a wrong version or these data are not reported. Furthermore, the figures reported in the authors' response lack SD or SE, in addition to lacking statistical analysis. Alternatively, the authors could report them in the table.

Since the authors report not finding differences in biochemical values, they could report it clearly in the M&Ms. I think it would be more correct to discuss the data separately anyway.

4) This does not mean that taste preferences were analyzed, but that they asked about food preferences and the two things do not coincide. In fact, the authors do not consider that in the choice of food an important role is played by the olfactory component, the main responsible for the hedonistic value.

5) at lines 106-107 the authors write the following “Patients were instructed to keep a food diary with daily recordings of the number of meals and the type of food consumed.” It doesn't seem to me that this is about quantity.

7) I still think that the analyses should consider age and sex separately and the conclusions should be drawn accordingly.

Round 3

Reviewer 4 Report

Comments and Suggestions for Authors

No comment

Author Response

Dear reviewer, thank you very much for the critical notes that helped improve our article!